**Data Availability Statement:** All relevant data are within the paper and its Supporting information files. Additional python script and sample data are

# Action potential metrics and automated data analysis pipeline for cardiotoxicity testing using optically mapped hiPSC-derived 3D cardiac microtissues

Arvin H. Soepriatna[1], Allison Navarrete-Welton[2], Tae Yun Kim[2], Mark C. Daley[1], Peter Bronk[2], Celinda M. Kofron[1], Ulrike Mende[2], Kareen L. K. Coulombe[1], Bum-Rak Choi[2]*

**1** Center for Biomedical Engineering, School of Engineering, Brown University, Providence, Rhode Island, United States of America, **2** Cardiovascular Research Center, Cardiovascular Institute, Rhode Island Hospital and Alpert Medical School of Brown University, Providence, Rhode Island, United States of America

☯ These authors contributed equally to this work.

* Bum-Rak_Choi@brown.edu

## Abstract

Recent advances in human induced pluripotent stem cell (hiPSC)-derived cardiac microtissues provide a unique opportunity for cardiotoxic assessment of pharmaceutical and environmental compounds. Here, we developed a series of automated data processing algorithms to assess changes in action potential (AP) properties for cardiotoxicity testing in 3D engineered cardiac microtissues generated from hiPSC-derived cardiomyocytes (hiPSC-CMs). Purified hiPSC-CMs were mixed with 5–25% human cardiac fibroblasts (hCFs) under scaffold-free conditions and allowed to self-assemble into 3D spherical microtissues in 35-microwell agarose gels. Optical mapping was performed to quantify electrophysiological changes. To increase throughput, AP traces from 4x4 cardiac microtissues were simultaneously acquired with a voltage sensitive dye and a CMOS camera. Individual microtissues showing APs were identified using automated thresholding after Fourier transforming traces. An asymmetric least squares method was used to correct non-uniform background and baseline drift, and the fluorescence was normalized ($\Delta F/F_0$). Bilateral filtering was applied to preserve the sharpness of the AP upstroke. AP shape changes under selective ion channel block were characterized using AP metrics including stimulation delay, rise time of AP upstroke, $APD_{30}$, $APD_{50}$, $APD_{80}$, $APD_{mxr}$ (maximum rate change of repolarization), and AP triangulation ($APD_{tri} = APD_{mxr} - APD_{50}$). We also characterized changes in AP metrics under various ion channel block conditions with multi-class logistic regression and feature extraction using principal component analysis of human AP computer simulations. Simulation results were validated experimentally with selective pharmacological ion channel blockers. In conclusion, this simple and robust automated data analysis pipeline for evaluating key AP metrics provides an excellent *in vitro* cardiotoxicity testing platform for a wide range of environmental and pharmaceutical compounds.

available from GitHub (https://github.com/arvinsoepriatna/AP_Analysis_Routines_Cardiotoxicity_Microtissues).

**Funding:** This study was supported by the National Heart, Lung, and Blood Institute at the National Institutes of Health U01 ES028184 to CRC, KLKC and UM, Brown Biomedical Innovation and Impact to Impact award to KLKC and BRC, and T35 HL094308 to ANW. The funders had no role in study design, data collection and analysis, decision to publish, or preparation of the manuscript.

**Competing interests:** The authors have declared that no competing interests exist.

**Abbreviations:** AP, Action potential; APD, Action potential duration; $APD_{30}$, $APD_{50}$, $APD_{80}$, APD measured at 30, 50, 80% repolarization; $APD_{MXR}$, APD measured from maximum rate repolarization using either $2^{nd}$ derivative or moving average subtraction; $APD_{tri}$, Triangulation of APD defined by $APD_{MXR} - APD_{50}$; CM, Cardiomyocyte; EAD, early afterdepolarization; $g_{K1}$, Conductance of inwardly rectifying potassium channel; $g_{Kr}$, Conductance of rapidly activating delayed rectifier potassium channel; $g_{Ks}$, Conductance of slowly activating delayed rectifier potassium channel; $g_{Na}$, Conductance of sodium channel; $g_{NaL}$, Conductance of late sodium channel; $g_{to}$, Conductance of transient outward potassium channel; hERG, human ether-à-go-go-related gene; hiPSC, human induced pluripotent stem cell; hiPSC-CM, hiPSC-derived cardiomyocyte; $I_{Ca}$, L-type calcium current; $I_{K1}$, Inwardly rectifying potassium current; $I_{Kr}$, Rapidly activating delayed rectifier potassium current; $I_{Ks}$, Slowly activating delayed rectifier potassium current; $I_{Na}$, Sodium current; $I_{NaL}$, Late sodium current; $I_{to}$, Transient outward potassium current; MA, Moving average; MAS, Moving average subtraction from the original trace; $p_{Ca}$, Permeability of L-type calcium channel; PCA, Principal component analysis; TdP, Torsades de Pointes; TTX, Tetrodotoxin.

## Introduction

Drug-induced cardiac toxicity, including arrhythmias and heart failure, is a growing challenge for physicians, regulatory agencies, and the pharmaceutical industry [1–5]. Notably, in the last decade, drug-induced QT prolongation leading to Torsades de Pointes (TdP) was the single most common cause of withdrawal or restriction of previously marketed drugs [1, 6–11]. More than 50 commonly prescribed medications can lead to QT prolongation [2, 12, 13], resulting in cardiac arrhythmias and sudden cardiac death. A drug that results in a corrected QT (QTc) interval >500 ms, or an increase of QTc >60 ms from baseline levels, is classified as a high risk drug for inducing TdP [14]. HERG (human ether-à-go-go-related gene) channel block is a main cause of drug-induced QT prolongation that may lead to early afterdepolarization (EAD) and the often fatal TdP. The hERG channels that underlie the rapidly activating delayed rectifier potassium ($K^+$) current ($I_{Kr}$) are blocked by a wide range of drugs due to the hydrophobic amino acids in the selective filter region of the pore [15–18].

QT prolongation by hERG channel block only presents one mechanism for cardiac arrhythmia initiation because proarrhythmic toxicity can also include bradyarrhythmia, slow conduction, and reentry leading to atrial and/or ventricular arrhythmias and Brugada syndrome [19–21]. Since the cardiac action potential (AP) is regulated by multiple ion channels, any drug that alters the kinetics of one or more of these ion channels/currents has the potential to initiate an arrhythmogenic event [22–24]. Conversely, drugs that nonspecifically block late $Na^+$ or $Ca^{2+}$ channels at an equal or greater extent than hERG channel block present lower risks in initiating arrhythmias than drugs that specifically block hERG alone [25], indicating the need to identify all drug targets to evaluate true arrhythmic risks [26]. Therefore, to better predict arrhythmia-related cardiotoxicity, it is important to rely on multiple sensitive metrics that capture changes during all phases of the AP, thereby reflecting summative alterations across multiple ion channels.

Cardiotoxicity testing in animal models often fails to recapitulate human responses to drugs due to species-specific differences in cardiac physiology, including differential electrophysiological, metabolic, and structural adaptations. Recent advances in human induced pluripotent stem cell (hiPSC) technology, however, have made the creation of an *in vitro* testing platform for cardiotoxicity assessment in human physiology possible [27–29]. Fluorescence imaging is an established method to record APs from hiPSC-derived cardiomyocytes [30–32] and microtissues [33–35]. Unlike extracellular potential measurements, fluorescence imaging allows researchers to track detailed voltage changes, equivalent to microelectrode recordings. Despite these advantages, fluorescence imaging has some disadvantages that must be mitigated experimentally and through computational approaches.

Several software packages are available for analyzing optical voltage image data of intact hearts or tissues [36–39]. These packages are optimized for impulse propagation, dispersion of AP duration (APD), or calcium transient dynamics from the surface of intact hearts to investigate triggered activity, alternans, conduction block, and reentry formation. However, optical mapping of hiPSC cardiac microtissues for cardiotoxicity testing encounters unique hurdles and thus requires different types of data analyses to accurately identify changes in AP shape associated with arrhythmia risks. Imaging of microtissues can be accelerated by recording APs from multiple samples simultaneously unlike whole heart mapping. Higher spatial resolution for voltage imaging of microtissues necessitates intense excitation light that can cause dye bleaching and a loss of fluorescence intensity within the recording time. Slight motion or changes to the fluid level in the recording chamber can alter baseline traces, complicating repolarization phase detection. Additionally, attempts to improve signal-to-noise ratio through over-smoothing can conceal rapid voltage response such as AP upstrokes [40], which require

caution when comparing data between different times or batches. Therefore, in this study, we developed an automated signal processing algorithm that overcomes these limitations to reliably detect drug-induced alterations in AP shape from fluorescent recordings.

We previously reported that hiPSC-derived cardiomyocytes (CMs) and fibroblasts (CFs) can co-self-assemble into 3D spherical microtissues that can be used as a cardiotoxicity testing platform [35, 41]. Although co-culture of CM and CF can facilitate compaction, CF percentage may alter the electrophysiology of CMs. Therefore, in this study, we examined how AP characteristics change with increase in CF content. We developed an automated data analysis pipeline that is computationally simple and robust for optical imaging datasets of 3D cardiac microtissues acquired with a voltage-sensitive dye. Advanced signal processing routines were applied to overcome baseline drift while minimizing distortion from the filtering of fluorescence voltage recordings. We identified 8 key metrics that comprehensively capture changes in AP shape in response to specific and nonspecific ion channel blockers. Computational simulations demonstrated that these AP metrics are sufficient to differentiate which ion channels are blocked, as supported by principal component analysis (PCA), and enable accurate predictions of ion channel blocks, as shown by logistic regression. Taken together, this study shows that automated data analysis of AP recordings from 3D cardiac microtissues provides a simple and robust *in vitro* cardiotoxicity testing platform that could be leveraged to establish safe human exposure levels for environmental and pharmaceutical compounds.

## Methods

### Cardiomyocyte differentiation

As described previously [42], we differentiated ventricular cardiomyocytes (CMs) from human induced pluripotent stem cells (WTC human male iPSCs, Gladstone Institutes) in cardiac differentiation media 3 (CDM3) by modulating the Wnt signaling pathway. Briefly, high-density hiPSC monolayer cultures (80–90%) were treated with a Wnt activator that inhibits glycogen synthase kinase 3 (4–5 µM Chiron 99021, Tocris) for 24 ± 1 hour, followed by a chemical Wnt inhibitor (5 µM IWP2, Tocris) at day 3. Differentiated cardiomyocytes began to beat between days 8 and 12, and once beating, were cultured in RPMI 1640 medium with B27 supplement (RPMI+B27; Gibco). Cardiomyocytes were then harvested and replated onto plates coated with Matrigel (Corning) for metabolic-based lactate purification [43]. Leveraging their unique ability to use lactate as an energy source, CMs were cultured in 4 mM sodium L-lactate (Sigma) in sodium pyruvate- and glucose-free DMEM for 4 days, effectively reducing non-cardiomyocyte population to improve cardiomyocyte purity.

### Generation of 3D cardiac microtissue

Sterilized 2% (wt/vol) agarose was pipetted into molds with 35 rounded-end pegs, each with an 800-µm diameter (3D Petri Dish, MicroTissues). The agarose was allowed to gel at room temperature before the hydrogels were removed from the molds and transferred to a 24-well plate. Hydrogels were equilibrated in RPMI+B27 + 1% Pen/Strep (Sigma) for at least 1 hr at 37˚C in an incubator with 5% $CO_2$. To generate scaffold-free 3D cardiac microtissues, purified cardiomyocytes were incorporated with human cardiac fibroblasts (hCFs) at specified ratios (5%, 15%, and 25% of total cell count) before pipetting the cell mixture into the seeding chamber of the agarose hydrogels at a density of $3-5 \times 10^5$ cells per hydrogel (~$8.5-14 \times 10^4$ cells per microwell). Cells were allowed to settle into the 35 recesses for 30 min before adding culture medium containing RPMI+B27 with 1% Pen/Strep, 10% FBS, and 5 µM Rho-associated kinase inhibitor (ROCK) inhibitor (Y27632, Sigma). Cell-containing hydrogels were field stimulated during culture with a 1 Hz, 10.0 V, 4.0 ms duration bipolar pulse train with an Ionoptix

C-Pace EP (Ionoptix, Westwood, MA) to facilitate electromechanical maturation. Within the first 3 days, the cell mixture compacted into 3D spherical cardiac microtissues, as the non-adhesive agarose hydrogel with hemispherical bottoms helped to guide cellular self-assembly. The culture medium was changed every two days, and cells were cultured under electrical stimulation for 6–8 days prior to optical mapping.

## Optical mapping of action potentials from 3D microtissues

Fig 1A shows a schematic of the optical mapping apparatus. Microtissues were mounted in a temperature regulated chamber (35˚C, dual temperature controller, WPI) and perfused with Tyrode's solution at 3 mL/min containing (in mM) 140 NaCl, 5.1 KCl, 1 $MgCl_2$, 1 $CaCl_2$, 0.33 $NaH_2PO_4$, 5 HEPES, and 7.5 glucose. Microtissues were loaded with a voltage-sensitive dye, di-4-ANEPPS (5 μM for 1 min and washed), to enable recording of membrane potential ($V_m$). Two programmable syringe pumps (Aladdin programmable syringe pump, WPI) were configured to perfuse normal and drug solutions at 3 mL/min. The drugs tested include 1) tetrodotoxin (TTX, 1 μM), a selective $Na^+$ channel blocker ($IC_{50}$ = 1 μM [44]); 2) flecainide (20 μM), a nonspecific $Na^+$ ($IC_{50}$ = 7.4 ~ 345 μM, use-dependent [45]) and hERG ($IC_{50}$ = 4 μM [46]) channel blocker; 3) Nifedipine (2 μM), a selective L-type $Ca^{2+}$ channel blocker ($IC_{50}$ = 0.2 μM [47]); 4) E4031 (2 μM), a selective hERG channel blocker ($IC_{50}$ = 7.7 nM ~ 397 nM [48, 49]); 5) Chromanol (30 μM), $I_{Ks}$ ($IC_{50}$ = 2~6 μM [50]) and $I_{to}$ blocker ($IC_{50}$ = 24 μM [51]). The concentrations of ion channel blockers were chosen to be above their reported $IC_{50}$-values. Drug solutions were perfused for 15 min at 3 mL/min speed to ensure that targeted ion channels are blocked. The perfusion was stopped for 5 min for additional incubation of drugs and settling of the solution level before fluorescence imaging, totaling to 20 min of drug incubation. Microtissues were paced with 2 ms biphasic stimulation pulse at 10 V/cm strength using two linear platinum electrodes. Fluorescence images were acquired at 1000–2000 frames/s using a CMOS camera (Ultima-L, 100x100 pixels, Scimedia, Japan). The field of view was set to 5×5 $mm^2$

**Fig 1. Schematics of fluorescence recording apparatus and data processing.** (A) Optical apparatus to record action potential (AP) from hiPSC-derived cardiac microtissues. (B) Schematic of the data processing pipeline. A series of data processing steps were taken to measure AP metrics: ① AP recording. ② segmentation for identification of individual microtissues, ③ baseline correction to remove fluorescence changes due to dye bleaching or fluctuation of perfusion solution, ④ filtering to increase signal-to-noise ratio while preserving sharpness of AP upstroke, ⑤ calculation of first derivative and moving average subtraction to detect AP upstroke and repolarization, and ⑥ AP metric measurement.

(with 50×50 μm$^2$ pixel resolution) using a custom-built optical apparatus, which included a Navitar lens (25 mm f0.95), a dichroic beam splitter box (Newport), and a combination of refocusing achromatic lenses (100 and 75 mm focal length lenses, Newport Corporation). This magnification enabled simultaneous recording of a 4x4 array of microtissues per scan. APs at baseline and 20 mins following drug perfusion were recorded and analyzed for changes in AP characteristics. An overall outline of the data processing steps is shown in Fig 1B, with detailed explanation provided in the main text.

## Computer modeling of action potential changes under selective ion channel block

AP changes under selective ion channel block were modeled using the human cardiac AP model by O'Hara *et al.* [52]. We modeled 40,000 cells with a Gaussian distribution of six conductance parameters ($g_{Na}$/$g_{NaL}$, $g_{to}$, $p_{Ca}$, $g_{Kr}$, $g_{Ks}$, and $g_{K1}$). The original parameter, referenced in [52], was set as the mean value of the distribution with the width set at a standard deviation equal to 10% of APD, mimicking the normal QTc range from 360 to 440 ms [53, 54]. The conductances of these ionic currents were distributed independently across cells except for $I_{Na}$ and $I_{NaL}$. The variance in $g_{Na}$ and $g_{NaL}$ was concordant in each cell (i.e., a cell with a decreased $g_{Na}$ was set with the same proportional decrease in $g_{NaL}$). 20 APs were stimulated at 2 sec intervals, and six AP metrics were extracted from the penultimate AP using the methods described below. Cells that failed to exhibit a normal AP (i.e., failure to initiate or failure to repolarize) were excluded from the analysis. To model the effects of pharmacological treatments that target different ion channels, six different ion channel blocks were modeled by reducing the relevant parameter ($g_{Na}$/$g_{NaL}$, $g_{to}$, $p_{Ca}$, $g_{Kr}$, $g_{Ks}$, and $g_{K1}$) by 25–100% at 25% intervals. The automated data analysis pipeline described below was used to extract six AP metrics (rise time, APD$_{30}$, APD$_{50}$, APD$_{80}$, APD$_{MXR}$, APD$_{tri}$). The stimulation delay metric was excluded in the analysis because AP stimulation was modeled by directly increasing the voltage in each cell, which always produces a delay equal to 0.

## Multi-class logistic regression and principal components analysis

Two methods were used to validate the ability of the metrics to differentiate between different ion channel blocks. First, principal components analysis (PCA) was conducted for both simulation and experimental data to determine whether AP metrics provided sufficient information to distinguish the effect of different pharmacological treatments. For computer simulation, 6 AP metrics, excluding excitability and stimulation delay, were used for PCA. For experimental data, 7 AP metrics, excluding the excitability metric, were used for PCA because all microtissues were excitable after acute drug treatment. The PCA algorithm was implemented in Python 3.6 using the *sklearn.decomposition* library. Data was pre-scaled with the StandardScaler tool from the *sklearn.preprocessing* library. Next, multi-class logistic regression models were trained to use AP metrics to predict which ion channel was blocked or which pharmacological compound had been applied. The multi-class logistic regression model was implemented in Python 3.6 using the *sklearn* library. Models were fitted to experimental and simulation data separately. In each case, 80% of the data were used for training and 20% were reserved for testing. The training set was scaled using the StandardScaler from the *sklearn.preprocessing* library, and the same scaling was applied to the test set. Hyperparameters were tuned on the training set using five-fold cross-validation. Since the experimental dataset was relatively small ($n = 136$), the model fitting, hyperparameter tuning, and testing process were repeated 100 times with different training/test splits to obtain the mean and standard deviation of the model accuracy. The results for the simulation dataset ($n = 199,983$) were more stable across different

training/test splits owing to the greater dataset size, so the model fitting, hyperparameter tuning, and testing process were only repeated 5 times to obtain the standard deviation.

## Python scripts of automated analysis

We have developed customized python scripts to automate signal processing and AP metrics measurement. The routines of segmentation, baseline subtraction, non-linear edge-preserving bilateral filtering, moving average subtraction, and measurements of AP rise time, $APD_{30}$, $APD_{50}$, $APD_{80}$, $APD_{mxr}$, $APD_{tri}$ are provided in the S1 Text.

## Statistical analyses

All data are expressed as mean ± standard deviation for $n$ microtissues unless otherwise indicated. Normality test was conducted with the Kolmogorov-Smirnov test, and statistical analyses were performed using Student's two tailed paired t-test comparing means of two measurements from the same microtissues before and after drug treatment. P-values < 0.05 were considered statistically significant.

## Results

### Changes in AP shape under specific ion channel block: Computer simulation study

Cardiac ventricular action potentials have 5 distinct phases: upstroke (phase 0), early repolarization (phase 1), plateau (phase 2), rapid repolarization (phase 3), and resting phase (phase 4). The openings and closings of multiple ion channels and their kinetics underly these unique AP phases, and each ionic current uniquely contributes to the different AP phases (Fig 2A). Our group previously suggested measuring 8 AP metrics (Fig 2B) to report changes across the different AP phases [35], with excitability defined as the percentage of successful AP formation in response to a pacing stimulus. For the computer simulation study, we used 6 AP metrics, excluding the stimulation delay and excitability metrics, because APs were always induced by stimulation current without delay and excitability was 100%. Computer simulation indicates that blocking the $Na^+$ current ($I_{Na}$), modeled by reducing channel conductivity, mainly prolonged the rise time of AP upstroke (shown in Fig 2C for $g_{Na}$ at 50%). Blocking the L-type $Ca^{2+}$ channel ($I_{Ca}$), shortened the late repolarization phase, decreasing both $APD_{50}$ and $APD_{80}$ to the same degree (Fig 2C for $p_{Ca}$ at 50%). Blocking the rapidly activating delayed $K^+$ current ($I_{Kr}$) resulted in a significant delay in the late repolarization phase, increasing $APD_{tri}$ significantly [35], while blocking the slowly activating delayed $K^+$ current ($I_{Ks}$) only induced a small delay during the late repolarizing phase (Fig 2C for gKr and gKs at 50%, mean and standard deviation in Table 1, and S1 Fig in S1 Text).

Fig 2D–2F shows a summary of AP metric changes under $I_{Na}$ block (purple, panel D), $I_{Ca}$ block (orange, panel E), and $I_{Kr}$ block (blue, panel F). PCA (panel G) was calculated from the percent change in each of the six metrics when each ion channel conductance was reduced to 25~75% of its original value. Two PCA axes were found to be sufficient to explain 97% of the variance in the dataset. The linear combinations of the metrics and the corresponding explained variance ratios are shown in Table 2. The dominant variable combinations in the PCA axes reflected APD prolongation (reflected by positive values for the $APD_{30}$, $APD_{50}$, $APD_{80}$, $APD_{mxr}$ and $APD_{tri}$ coefficients in axis 1) and a significant change in AP rise time (axis 2). These PCA axes were capable of clearly separating the 25% block of $I_{Kr}$ from other ion channel blocks, and the $I_{Ca}$ and $I_{Na}$ blocks were also mostly separable (Fig 2G). However, a 25% block of $I_{to}$ and $I_{Ks}$ did not significantly change AP metrics, and these PCA axes did not

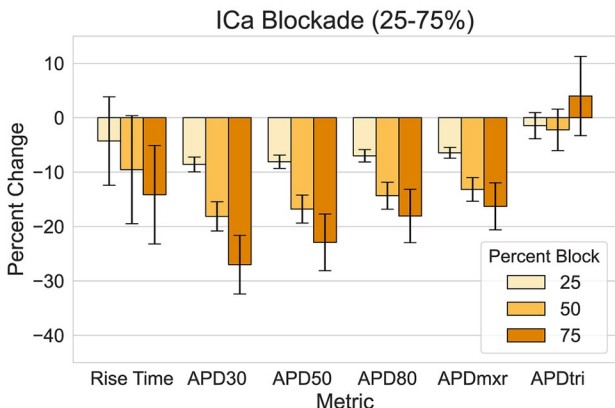

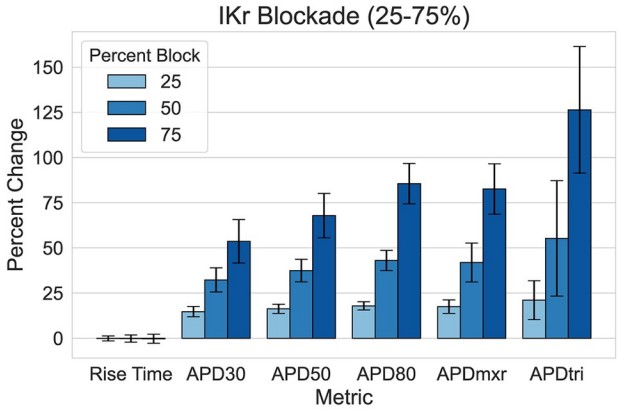

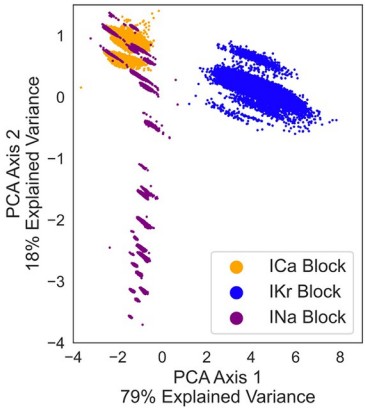

**Fig 2. Selection of action potential metrics to detect ion channel block through computer modeling.** (A) Human action potential model by O'Hara et al. [52] and the underlying major ionic currents: $Na^+$ ($I_{Na}$), transient outward $K^+$ ($I_{to}$), $Ca^{2+}$ ($I_{Ca}$), rapidly ($I_{Kr}$) and slowly ($I_{Ks}$) activating delayed rectifier $K^+$ currents, and inward rectifier $K^+$ current ($I_{K1}$). (B) Illustration of AP metrics to detect AP shape changes by ion channel block; AP upstroke rise time, plateau phase ($APD_{30}$ and $APD_{50}$), and phase 3 repolarization ($APD_{80}$, $APD_{mxr}$, and $APD_{tri} = APD_{mxr}-APD_{50}$). (C) Modeling of AP changes in response to specific block of indicated ion channels to 50% conductance (gX 50%). $I_{Na}$ block exclusively alters AP upstroke; $I_{Ca}$ block shortens APD by depressing the plateau $V_m$ which accelerates repolarization; $I_{Kr}$ block slows the phase 3 repolarization; and $I_{Ks}$ block slightly prolongs APD. (D-F) Simulated changes in AP metrics following specific blocking of three major ionic currents ($I_{Na}$, $I_{Ca}$, and $I_{Kr}$) at 25%, 50% and 75%. Details of the modeling and parameter sets are described in Methods. (G) PCA plot of $I_{Na}$, $I_{Ca}$, $I_{Kr}$ block. Feature extraction was done using principal component analysis (PCA). Distributions of AP metric changes are shown for $I_{Na}$ (purple), $I_{Ca}$ (orange), and $I_{Kr}$ (blue) at 75% block. PCA indicates two major features that best distinguish AP metric changes between the five simulated blocks ($I_{Na}$, $I_{Ca}$, $I_{Kr}$, $I_{to}$, and $I_{Ks}$; see Tables 1 and 2).

**Table 1. AP metric changes under specific ion channel block in computer simulations.** Results are reported as mean +/- standard deviation (percent change of mean from cell mean without channel block). 40,000 single cells with channel conductances varied to reflect population-level heterogeneity were paced at 0.5 Hz as described in Methods. AP metrics were determined for the penultimate beat.

| | $APD_{30}$ (ms) | $APD_{50}$ (ms) | $APD_{80}$ (ms) | $APD_{mxr}$ (ms) | $APD_{tri}$ (ms) | Rise Time (ms) |
|---|---|---|---|---|---|---|
| No Block | 188 +/- 22 | 230 +/- 27 | 272 +/- 33 | 307 +/- 40 | 84 +/- 13 | 4.75 +/- 0.49 |
| $I_{Na}$ Block (25% $g_{Na}$, 25% $g_{NaL}$) | 194 +/- 41 (+3%) | 230 +/- 45 (0%) | 273 +/- 49 (0%) | 300 +/- 94 (-2%) | 79 +/- 12 (-6%) | 8.57 +/- 2.58 (+ 80%) |
| $I_{to}$ Block (25% $g_{to}$) | 181 +/- 22 (-4%) | 227 +/- 27 (-1%) | 269 +/- 33 (-1%) | 305 +/- 40 (-1%) | 88 +/- 13 (+5%) | 4.87 +/- 0.44 (+ 3%) |
| $I_{Ca}$ Block (25% $p_{Ca}$) | 138 +/- 21 (-27%) | 178 +/- 24 (-23%) | 223 +/- 29 (-18%) | 257 +/- 29 (-16%) | 85 +/- 11 (+1%) | 4.04 +/- 0.19 (- 15%) |
| $I_{Kr}$ Block (25% $g_{Kr}$) | 287 +/- 20 (+53%) | 384 +/- 27 (+67%) | 501 +/- 38 (+84%) | 557 +/- 47 (+81%) | 214 +/- 25 (+155%) | 4.74 +/- 0.50 (-0.3%) |
| $I_{Ks}$ Block (25% $g_{Ks}$) | 198 +/- 25 (+5%) | 242 +/-31 (+5%) | 285 +/- 38 (+5%) | 320 +/- 40 (+4%) | 87 +/- 14 (+4%) | 4.75 +/- 0.49 (0%) |

clearly separate them from the $I_{Na}$ case (S1 Fig in S1 Text). Taken together, these simulation results indicate that AP metrics of rise time, $APD_{30}$, $APD_{50}$, $APD_{80}$, $APD_{mxr}$, and $APD_{tri}$ are sufficient to detect key ion channel blocks that can increase the risks for cardiac arrhythmias.

## Effects of hCF content on 3D microtissue compaction rates and electrophysiology

Previous studies [35, 55–58] showed that co-culture of CM and CF improves the formation of 3D microtissues with gene expression profiles that can closely mimic the native myocardium. However, depending on the CF content and their activation state, inclusion of CF can alter AP dynamics such as APD, conduction, and automaticity [59, 60]. Therefore, we investigated the effect of CF content on AP characteristics. Human CFs (hCF; passage number < 5) were incorporated at ratios of 5%, 15%, and 25%, and microtissue compaction and APs were characterized. hiPSC-CM and hCF self-assembled within a day after being seeded into agarose hydrogels, and compaction increased to form 3D spheroid structure by day 7 (Fig 3A–3C). AP metrics (Fig 3D and 3E) show that up to 25% of hCF inclusion did not induce major changes in excitability or AP characteristics, although we observed a trend of decreasing $APD_{30}$ with increasing hCF content, which may indicate a potential effect of hCF on phase 1 of the AP. Since 5% of hCF is sufficient to facilitate compaction of 3D spheroids, we used 5% hCF content for subsequent drug testing.

## Image segmentation of fluorescence recordings

Serial recordings of APs from individual microtissue can be time consuming to acquire, especially when APs from multiple microtissues (35 microtissues in our high-throughput platform) need to be collected in drug-dose response experiments. Furthermore, degraded image/signal quality is a significant concern due to dye bleaching and phototoxic damage from the intense excitation light during prolonged imaging. In our platform, $4 \times 4 = 16$ microtissues were imaged simultaneously, reducing overall recording time. APs from each microtissue can then

**Table 2. Principal component analysis of specific ion channel block in computer modeling.** PCA1 is a major axis that underlies overall APD prolongation and explains 79% of the variance in the simulation data. PCA2 mainly detects changes in the rise time of the AP upstroke with a smaller contribution due to shortening of $APD_{30}$ and prolongation of APD triangulation. In computer simulation, the delay between stimulation and AP upstroke was small compared to the sampling rate and the stimulation delay parameter was not incorporated in calculating PCA axes.

| PCA Axis | $APD_{30}$ | $APD_{50}$ | $APD_{80}$ | $APD_{mxr}$ | $APD_{tri}$ | Rise Time | Stim Delay | Explained Variance Ratio |
|---|---|---|---|---|---|---|---|---|
| PCA1 | 0.44 | 0.45 | 0.46 | 0.46 | 0.43 | -0.05 | Not included | 0.79 |
| PCA2 | -0.20 | -0.09 | -0.04 | 0.02 | 0.21 | -0.95 | Not included | 0.18 |
| PCA3 | -0.47 | -0.29 | -0.08 | 0.13 | 0.77 | 0.30 | Not included | 0.03 |

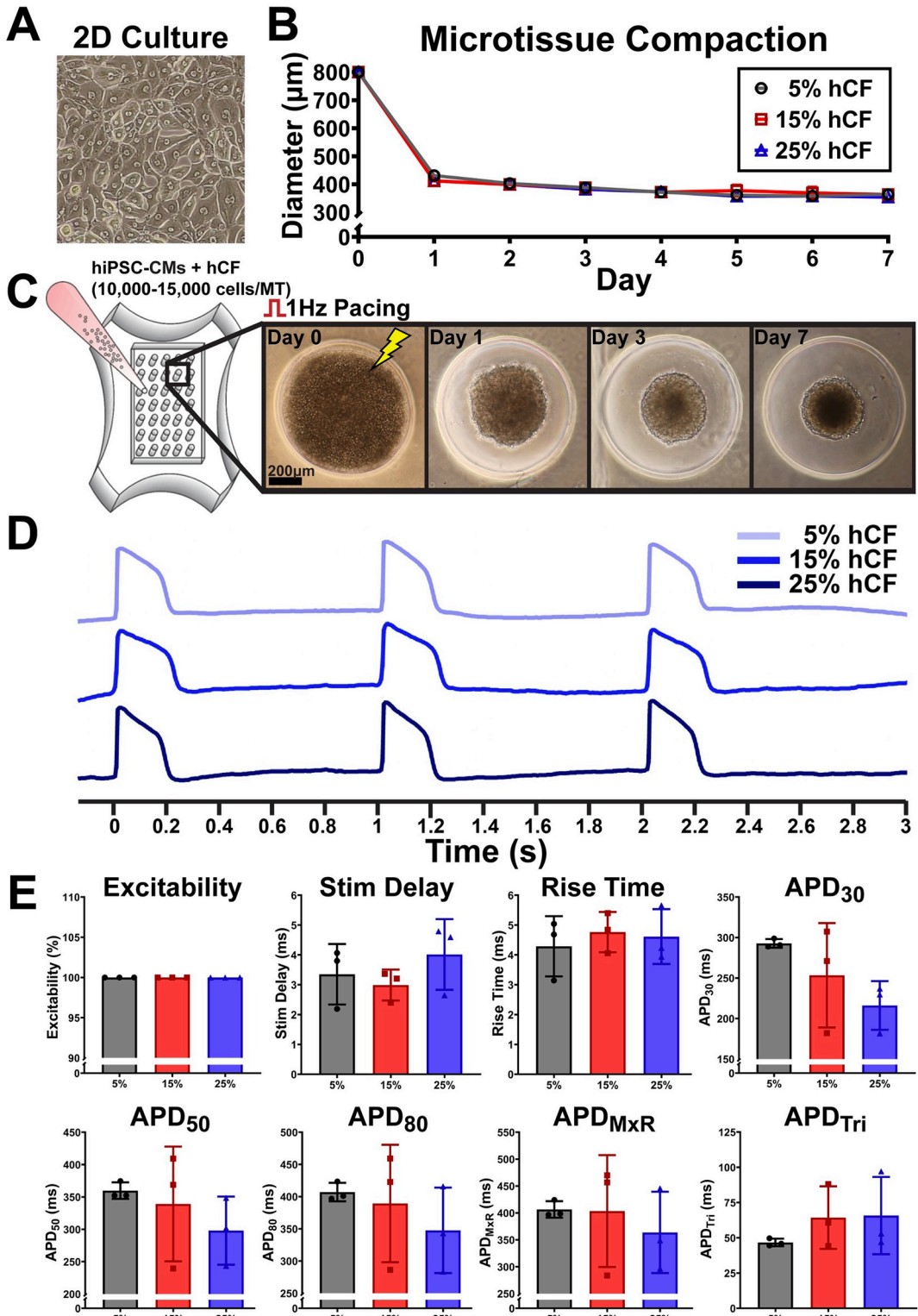

**Fig 3. Effects of human cardiac fibroblast (hCF) percentage on microtissue compaction and characterization of AP.** (A) Phase-contrast image of 2D culture of purified hiPSC-CMs for microtissue generation. (B) Microtissue compaction assessed by reduction in spheroidal microtissue diameter over 7 days of culture. (C) Sample images illustrating microtissue compaction over indicated time. (D) Sample action potential traces with indicated hCF content. (E) AP characteristics of microtissues with indicated hCF content recorded after 6–8 days in culture. Each dot represents averaged AP metrics from each mold (n = 3 molds, 32–35 microtissues per mold).

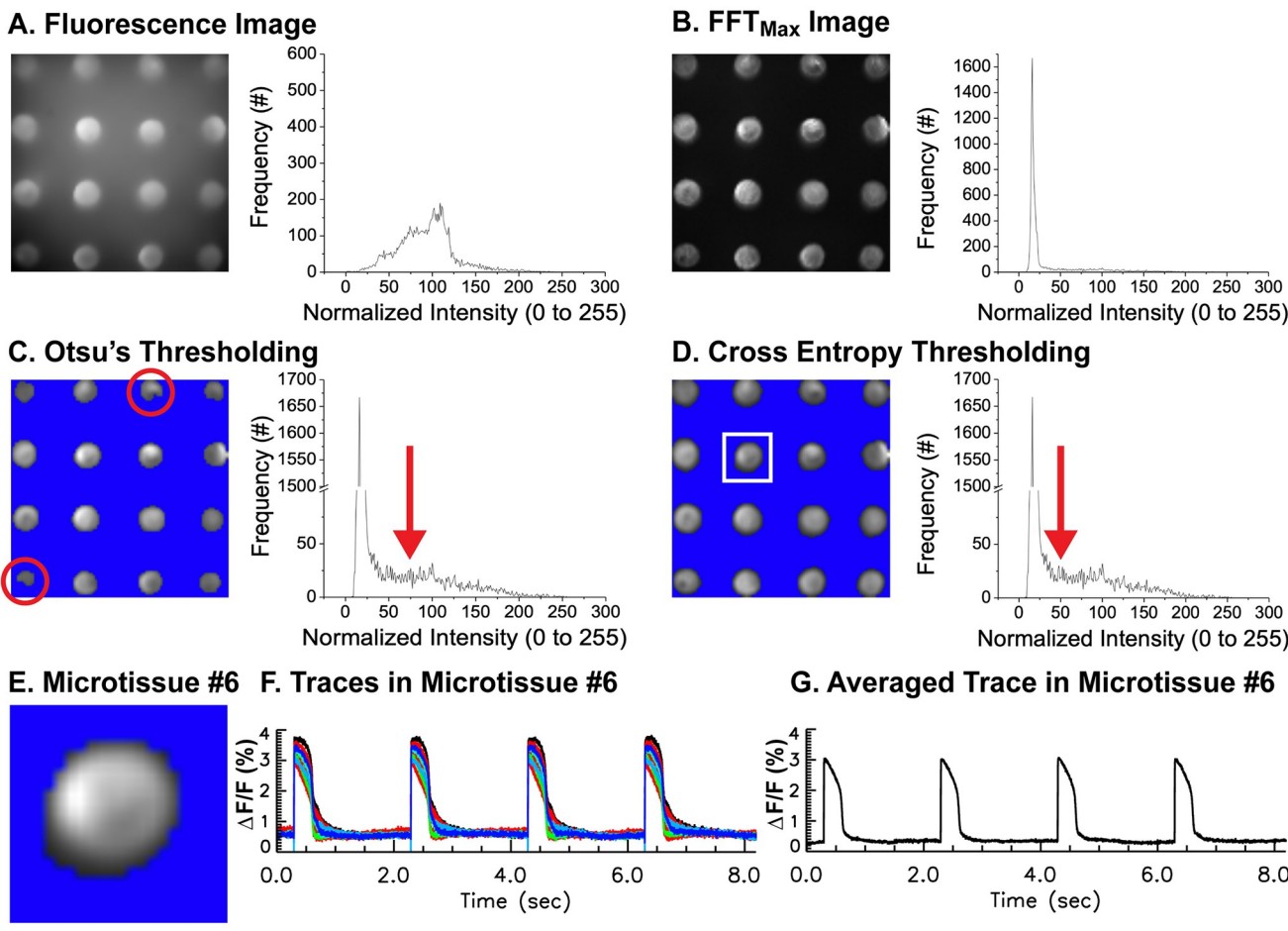

**Fig 4. Segmentation of individual microtissues from fluorescence voltage images.** (A) Sample fluorescence image (left) and a histogram of the sample image (right). Traces from each pixel were transformed to frequency domain using FFT, and the maximum FFT amplitude within pacing cycle length were calculated to reconstruct a $FFT_{max}$ image. (B) $FFT_{max}$ image that highlights regions showing full AP amplitude. The histogram of $FFT_{max}$ image (right panel) shows the best contrast image compared to fluorescence intensity in panel A. (C) Otsu's thresholding and corresponding histogram in log scale showed detailed distribution of pixels with lower intensity. Since Otsu's method selects a more conservative threshold value (red arrow), the segmentation often results in a smaller or distorted selection of microtissues (red circles). (D) Cross entropy thresholding via a minimum cross entropy algorithm improves the selection of pixels with AP signals (see the red arrow, mean threshold using cross entropy = 30 ± 17 vs. Otsu's method = 72 ± 16, n = 30 microtissues). After segmentation, a region-labeling algorithm identified individual microtissues in a solid circular shape compared to Otsu's method in panel C (mean area of microtissue detected using cross entropy = 160 ± 31 pixels vs. Otsu's method = 99 ± 11 pixels, n = 30 microtissues). (E) Sample selected microtissue marked with a white box in panel D. (F) Superimposed traces from all the pixels within microtissue #6 in panel E. (G) Averaged trace from all the pixels in panel F.

be individually analyzed by applying segmentation algorithms. Fig 4A shows a representative snapshot of the acquired fluorescence image. The histogram of fluorescence intensity shows a wide range of background and variation in dye staining, illumination of excitation light, and reflection from the mold, making it difficult to identify the proper threshold. Therefore, individual pixels were transformed into the frequency domain using FFT, and $FFT_{max}$ values in the range of pacing frequency were selected to reconstruct the image, which highlights pixels with proper APs. A representative $FFT_{max}$ image and its histogram in Fig 4B show higher contrast (entropy = 4.8 ± 0.2 vs 3.1 ± 0.7 for fluorescence and $FFT_{max}$ images respectively, *n* = 10 molds). Otsu's thresholding was then implemented to determine a threshold value that best reduces both the variances of pixel intensity in the background and the microtissues. Although this algorithm works well in identifying microtissues, the selected area is often smaller than the

microtissues (Fig 4C, red circles). Alternatively, a cross entropy algorithm that reduces entropy between the image and its segmented region can be used to improve the selection of pixels (Fig 4D). After segmentation, a region-labeling algorithm identified individual microtissues. The variation of APDs within the same microtissue was small, likely because electrical coupling between cells masks differences between individual cells within the same microtissue (standard deviation of APD = 1.6 ms, *n* = 34, see S2 Fig in S1 Text and S1 Movie). Once individual microtissues were identified (Fig 4E), fluorescence traces can be averaged to obtain a representative AP trace for each microtissue (Fig 4F and 4G).

## Removal of baseline drift and calculation of $\Delta F/F_0$

Although rare, fluorescence recordings can be affected by dye bleaching from intense excitation light and fluctuation in perfusion solution level, causing a baseline drift (Fig 5A). To adjust for this problem, we fitted a polynomial to the baseline using an asymmetric least squares regression [61]. Instead of the cost function of smoothness originally proposed in [61], we used polynomial fitting with both fast and slow frequency components to reduce the calculation time of matrix inversion and distortion of cardiac AP signals. Typically, a $3^{rd}$ or $5^{th}$ order polynomial was sufficient to fit the baseline of our fluorescence recordings. The principle of asymmetric least squares regression is based on assigning greater weight to points that are close to the baseline. Fig 5B–5J illustrates the iterative approach we used to find the baseline. First, AP traces were downsampled to 256 or 512 points (Fig 5B) to reduce calculation cost. The initial iteration fits the downsampled traces to the $3^{rd}$ order polynomials using a conventional least squares regression method (Fig 5C). In the second iteration, weights are assigned to each timepoint based on the value of the point with respect to the fitted polynomial from the first iteration. All data points that fall above the fit curve are weighted by a factor of 0.001. The curve is then fitted again to a new polynomial, now with a stronger contribution from the data points that fall below the fitted curve (Fig 5D and 5E). Typically, 3 to 5 iterations were sufficient to find a reasonably fitted curve to the baseline drift (Fig 5F–5I). The signal data was then normalized by dividing the difference from the baseline by the baseline value ($\Delta F/F_0$) at each timepoint (Fig 5J).

## Automated analysis for AP upstroke

After baseline correction, the traces were smoothed with bilateral filtering [62] to preserve the sharpness of AP upstrokes. In conventional Gaussian filtering, increasing the σ parameter reduces noise but significantly blurs and slows the upstroke of the AP (Fig 6A, top panel). Bilateral filtering [62] is an extension of Gaussian filtering that incorporates intensity-weighting to preserve sharpness of edges.

The intensity term in the bilateral filtering equations (see S1 Text) adjusts the weights of points depending on the similarity of intensity. If the neighboring point has an intensity similar to the reference point, it will be assigned a larger weight and contribute to the averaging step of the smoothing filter. Conversely, if the neighboring point has a markedly different intensity from the reference point, the weight is decreased so the point is effectively removed from the averaging step. As a result of this intensity-weighting, the rapid rise of the AP upstroke is still preserved after smoothing, and the results are relatively insensitive to the choice of σ and the window size (Fig 6A, middle and bottom panel), allowing accurate measurements of AP rise time. Additional details of bilateral filtering with varying parameters are available in the S3 Fig in S1 Text.

AP upstroke analysis requires accurate identification of the time of takeoff and the time of the AP peak. Traditional methods typically define takeoff as the time when the value is greater

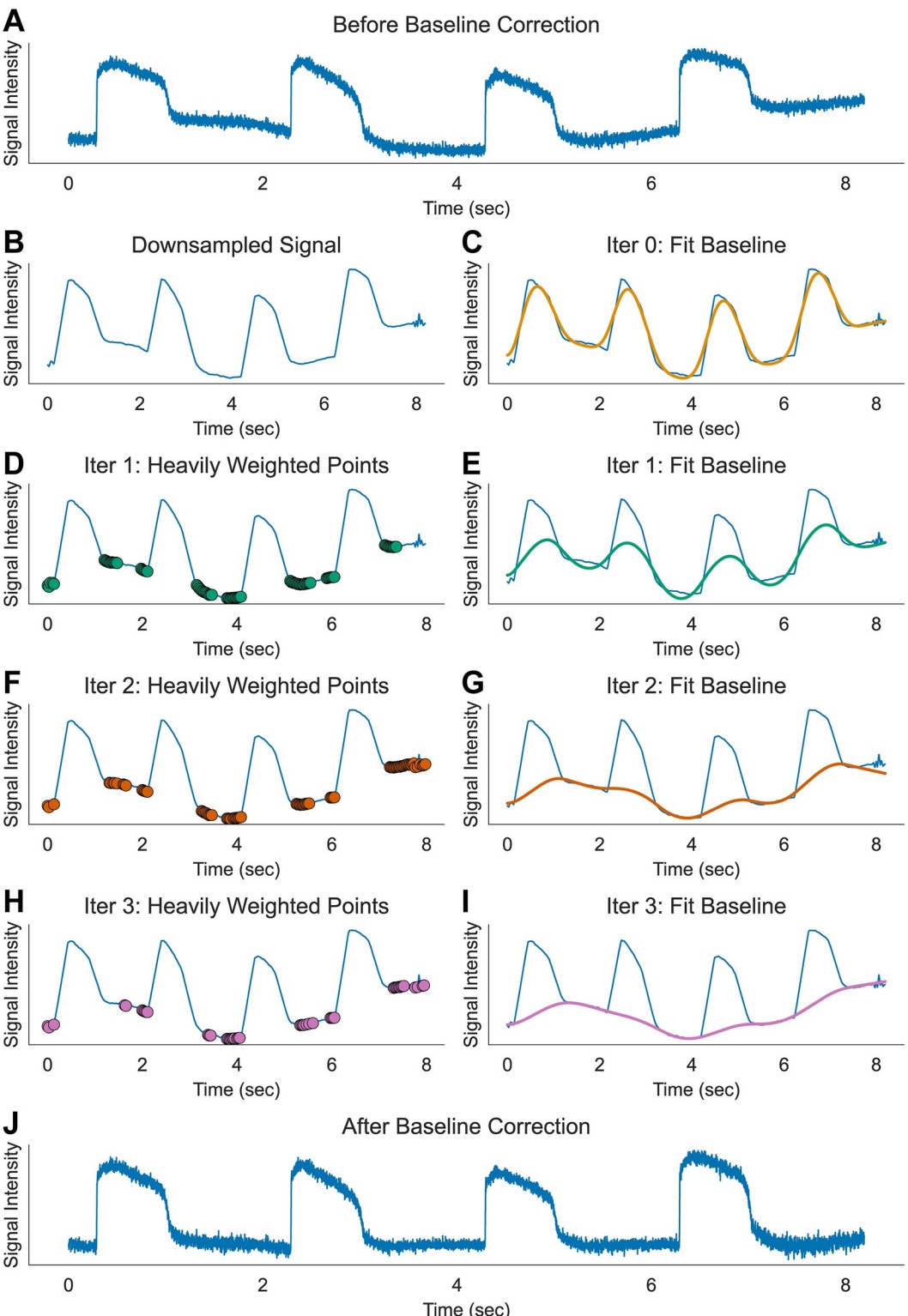

**Fig 5. Baseline correction.** (A) Sample raw trace of AP from single pixel with baseline drift before correction. Dye bleaching and fluctuation from perfusion flow may cause slow drift in baseline. (B-I) Iterative method of polynomial fitting for baseline estimation using asymmetric least squares method. Step 1: Resample the original signal to lower time resolution to accelerate calculation (here, 128 points are sampled from 8192 points, panel B). Step 2: Fit the signal with least squares polynomial fitting (her[e] 3rd order, panel C). Step 3: Apply reduced weight when the signal is above the polynomial fitting by multiplying

asymmetric parameter p (here 0.001, panel D). Step 4: Apply polynomial fitting again and repeat adjusting weight (panel E). Step 5: repeat iteration until the baseline estimation converges (panel F-I). (J) Sample trace after 5 iterations of baseline correction.

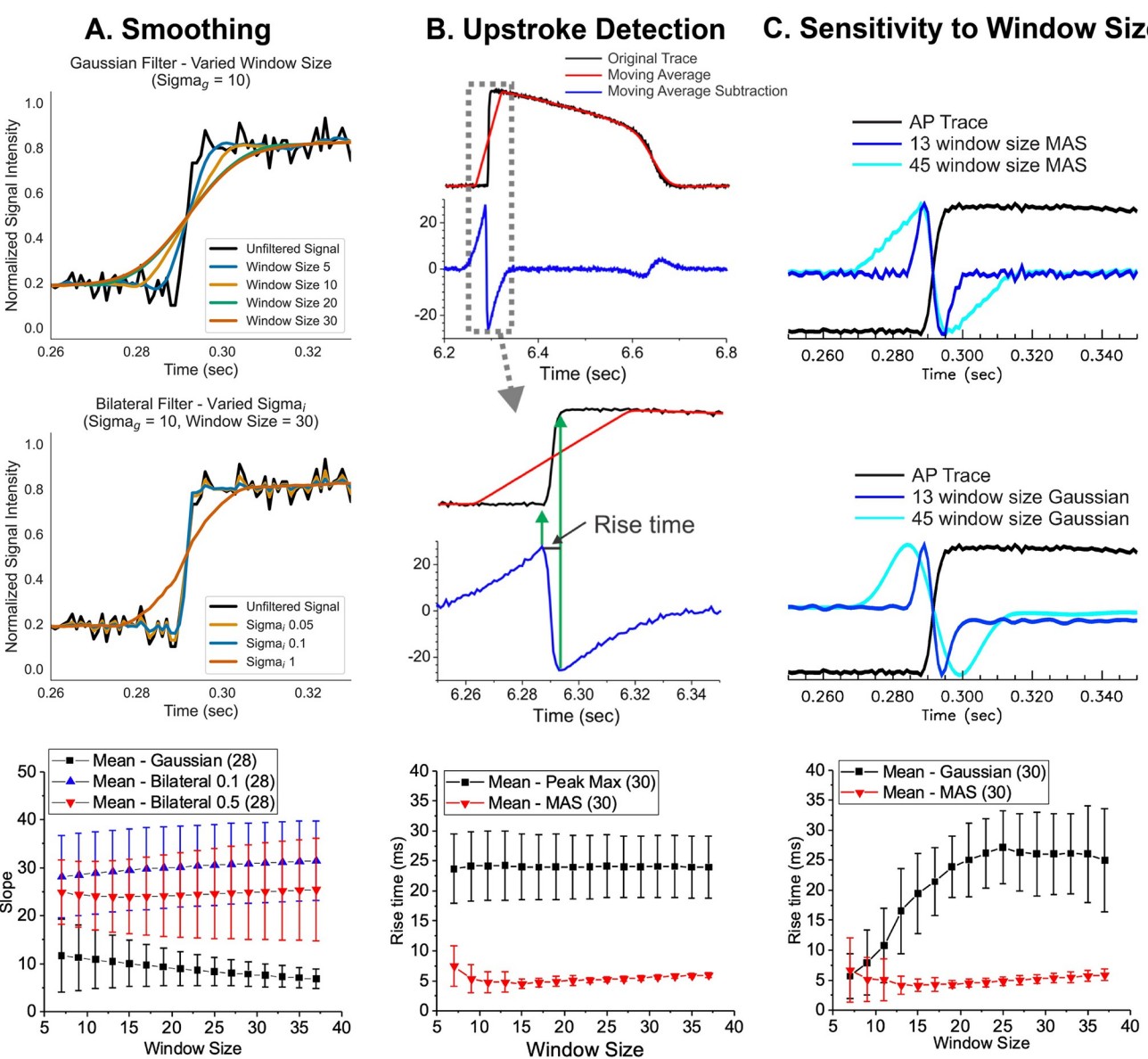

**Fig 6. AP upstroke analysis.** (A) Non-linear bilateral filter to preserve the sharpness of AP upstroke. Sample traces of AP before (black) and after applying Gaussian (top panel traces, blue to red) and bilateral (middle panel traces, blue to red) filters. AP upstroke is preserved after bilateral filtering even with a significantly larger window size (middle panel) compared to typical Gaussian filtering (top panel). The slope of AP upstroke was steeper with bilateral filter vs. gaussian filter (triangle for bilateral vs. black square plots for gaussian filter in the bottom panel), showing the advantage of edge-preserving bilateral filter to reduce noise in AP traces. (B) Upstroke detection using a moving average subtraction algorithm to detect the takeoff and peak time of AP upstroke. The moving average was calculated (red, window size = 45 points) and the original trace (black) was subtracted (blue), which detects a rapid change in AP upstroke. The middle traces show that the maximum of moving average subtraction (MAS) occurs at the initial rise of AP upstroke and the minimum of MAS occurs when the rapid rise of the AP upstroke is finished. The bottom panel compares the rise time of AP upstroke measured by time-to-max vs. MAS method. After sufficient filtering with 15-point bilateral filter, the standard deviation of the rise time measurements using MAS is significantly lower, increasing sensitivity to detect potential changes in AP upstroke by $Na^+$ channel blockers. (C) AP upstroke analysis by MAS compared to Gaussian $2^{nd}$ derivatives. The rise time measurement using MAS is less sensitive to the choice of window size (top), while the rise time measurement using Gaussian $2^{nd}$ derivatives (middle) is highly sensitive to the choice of window size (bottom).

than the background noise, with peak time defined as the time at which the maximum value is reached. This method often depends on high temporal resolution and a noise-free baseline and peak value. We instead developed a computationally simple algorithm to detect the initial takeoff and peak times using moving average subtraction (MAS). Fig 6B illustrates the calculation process. First, a simple moving average is calculated using a large window size (red trace in Fig 6B, window size = 45). The original trace is then subtracted from the moving average trace (blue trace). The middle panel shows that the maximum of MAS occurs at the initial rise of AP upstroke, and the minimum of MAS occurs when the rapid rise of AP upstroke is finished, enabling reliable rise time measurements. The rise time measurements using MAS shows significantly smaller values and variations, compared to the conventional rise time measurements using time-to-peak, because it does not solely rely on the time/location of potentially noisy AP peaks, which can shift significantly when a conventional gaussian filter approach is employed for noise removal. (Fig 6, bottom panel). Indeed, the MAS method is insensitive to the choice of window size (Fig 6C, top traces) compared to a similar method using a Gaussian $2^{nd}$ derivative (Fig 6C, middle traces), which shows significant increase of rise time measurements with smoothing window size (Fig 6C, bottom panel). The MAS method also reliably detects the AP rise when the resting membrane potential rises slowly, such as in the case of a pacemaker potential, in which a slowly rising membrane potential leads to spontaneous activity or automaticity (see S3F Fig in S1 Text). These results support the idea that the MAS method can be widely used in a variety of conditions to reliably measure AP takeoff and rise time.

## Automated analysis of repolarization

Accurate detection of cardiac AP repolarization in fluorescence recordings can be hampered by baseline drift and contraction artifacts. Previously, an edge detection method using $2^{nd}$ derivatives was proposed to detect the repolarization time associated with the peak of $I_{K1}$ during phase 3 repolarization in computer simulations [63]. We found that the time point of maximum MAS can also be used to detect repolarization in a similar fashion to the $2^{nd}$ derivative method (Fig 7A). As in AP upstroke analysis, the MAS method for repolarization time detection is computationally simple and relatively insensitive to the choice of σ and window size. Additional metrics such as $APD_{30}$, $APD_{50}$, and $APD_{80}$ were determined to be the earliest times after each AP peak where the signal value fell below the relevant fractions of the difference between the AP peak and the subsequent baseline (see Fig 7B). Both Gaussian $2^{nd}$ derivative method and MAS converges to the similar values with increased smoothing (Fig 7C), but the mean value of APD measured with the MAS method (red) was less affected by the choice of smoothing window size (Fig 7C and 7D).

## Validation of AP metrics with multi-class logistic regression and PCA analysis applied to experimental data

We applied the automated segmentation and signal processing algorithms to experimental data to measure changes in AP metrics under selective ion channel block. hiPSC-derived 3D cardiac microtissues were generated, and Aps were measured in the absence and presence of known selective ion channel blockers. Tetrodotoxin (TTX), which selectively blocks $Na^+$ channels, increased rise time and stimulation delay, defined as the delay between the stimulus pulse and AP upstroke (Fig 8A). The class I antiarrhythmic drug flecainide increased both rise time and stimulation delay similar to TTX, indicating that flecainide blocks $Na^+$ channels (Fig 8B). However, flecainide also increased $APD_{30}$, $APD_{50}$, $APD_{80}$, $APD_{mxr}$, and $APD_{tri}$, supporting published work which showed that flecainide also blocks $I_{Kr}$ [64]. Nifedipine, which blocks L-

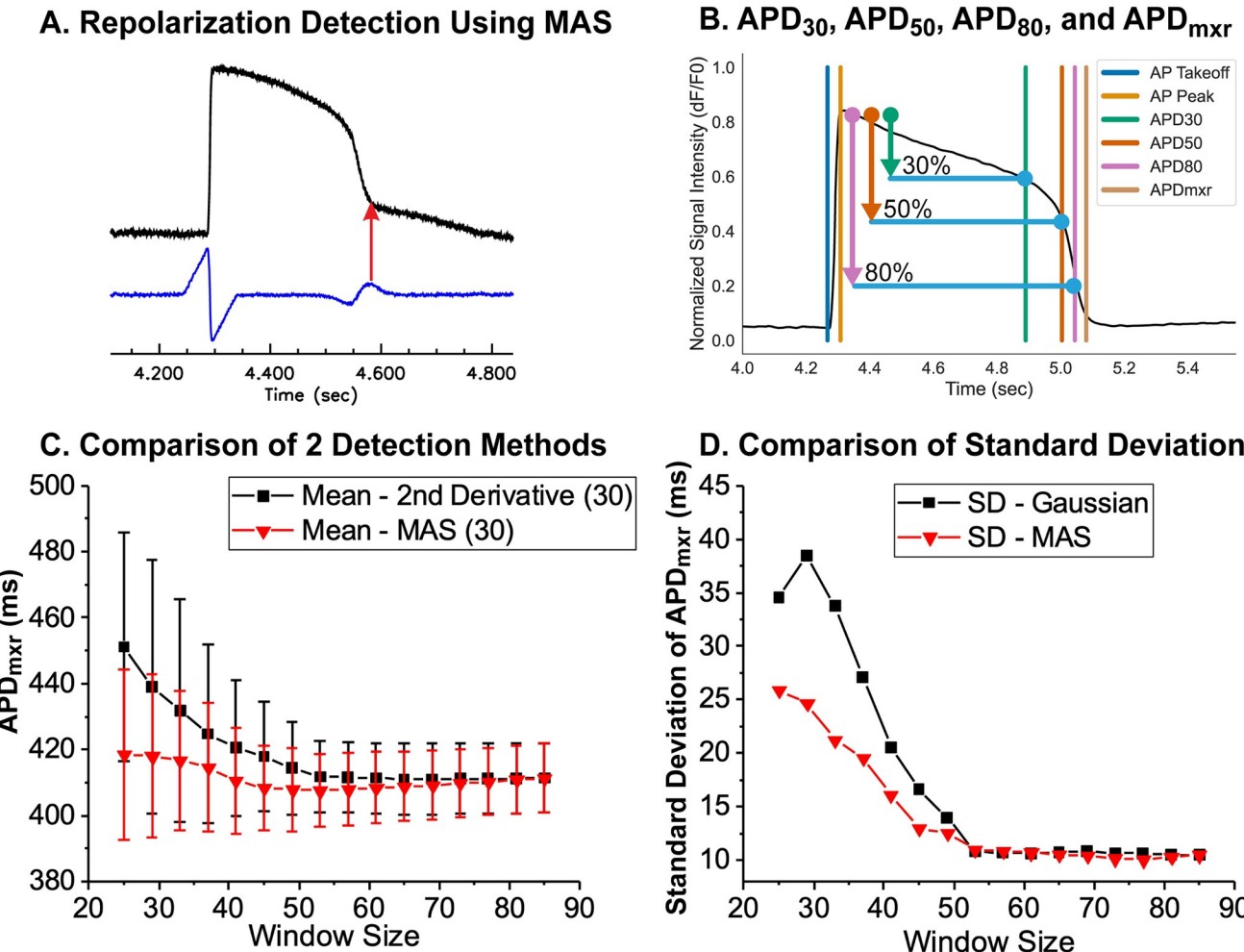

**Fig 7. Detection of AP repolarization.** (A) Repolarization time using moving average subtraction (MAS). MAS shows a secondary peak during the late phase of repolarization when the rapid repolarization slows down to reach the resting membrane potential. This time point is termed the maximum rate change of repolarization ($APD_{mxr}$). (B) Automatic detection of $APD_{30}$, $APD_{50}$, $APD_{80}$, $APD_{mxr}$ from a sample AP recorded from a single microtissue. (C) Comparison of APD measurements using MAS vs. 2nd derivative methods. A longer window size was required to measure APDs (>45) compared to AP upstroke analysis (~13, see Fig 6C). The mean value of APD using the MAS (red) method is less sensitive to the smoothing window size using the Gaussian 2nd derivative method (black). (D) Increasing the smoothing window size reduces the standard deviation of APD measurements. $N = 30$ microtissues.

type Ca channels, shortened $APD_{30}$, $APD_{50}$, $APD_{80}$, $APD_{mxr}$, and $APD_{tri}$ (Fig 8C). E4031, which selectively blocks $I_{Kr}$, prolonged $APD_{30}$, $APD_{50}$, $APD_{80}$, $APD_{mxr}$, and $APD_{tri}$. Importantly, E4031 significantly increases $APD_{tri}$ (Fig 8D) as predicted by computer simulations (Fig 2D, [35, 65, 66]) and triggered EADs in a subset (22%) of our microtissues. Finally, chromanol, which selectively blocks $I_{Ks}$ compared to $I_{Kr}$, slightly but significantly prolonged $APD_{30}$, $APD_{50}$, $APD_{80}$, and $APD_{mxr}$, without altering $APD_{tri}$ (S4 Fig in S1 Text), in agreement with our computational model that showed $I_{Ks}$ block resulted in a small delay of the late repolarizing phase (S1C Fig in S1 Text).

The observed changes in AP metrics under specific ion channel blockers agreed with the results from computer modeling (Fig 2 and S1 Fig in S1 Text). The plot of the experimental data in two PCA axes (Fig 8E, Table 3) were similar in nature to that obtained from the computer simulation result (Fig 2D, Table 2; see also S5 Fig in S1 Text for computer simulation

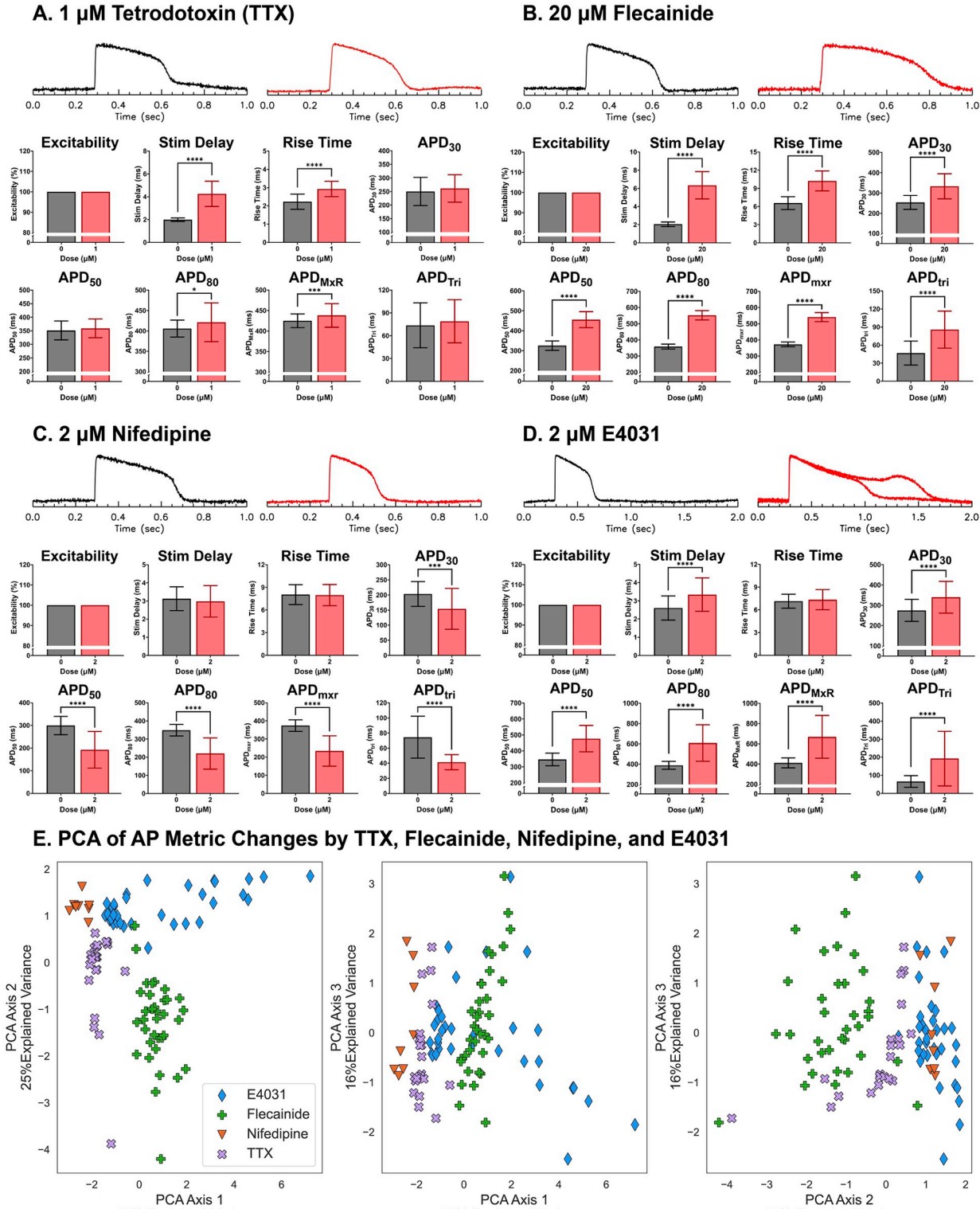

**Fig 8. Experimental validation of AP metric changes in response to known ion channel blockers.** Representative AP traces before and 15 to 30 mins after drug exposure at indicated concentration (top) and automatically quantified AP metrics from microtissues containing 5% hCF after 6~8 days in culture (bottom). (A) Tetrodotoxin (TTX, 1 μM), a selective Na⁺ channel blocker, increased stimulation delay and rise time and moderately increased $APD_{80}$ and $APD_{mxr}$ ($n = 32$ microtissues). (B) Flecainide (20 μM), a $I_{Na}$ and $I_{Kr}$ blocker, increased stimulation delay and rise time as well as $APD_{30}$, $APD_{50}$, $APD_{80}$, $APD_{mxr}$, and $APD_{tri}$ (n = 35 microtissues). The increase ion $APD_{tri}$ is caused by larger prolongation of

APD$_{mxr}$ compared APD$_{30}$ and APD$_{50}$. (C) Nifedipine (2 μM), a selective L-type Ca$^{2+}$ channel blocker, shortened APD$_{30}$, APD$_{50}$, APD$_{80}$, APD$_{mxr}$, and APD$_{tri}$ (n = 30 microtissues). (D) E4031 (2 μM), a selective I$_{Kr}$ blocker, increased APD$_{30}$, APD$_{50}$, APD$_{80}$, APD$_{mxr}$, and APD$_{tri}$ (n = 32 microtissues). Due to pronounced APD prolongation, the time scale is shown over 2 sec for E4031 traces. Additionally, two sample action potential traces are shown for E4301, one without EAD and another with EAD (n = 7/32 microtissues). (E) Discriminating ion channel blocks using AP metrics and PCA. Distribution of AP metric changes are shown for TTX (purple), flecainide (green), nifedipine (orange), and E4031 (blue). Principal component axes are shown in Table 3.

**Table 3. Principal components analysis of experimental data using TTX, Flecainide, Nifedipine, and E4031 in Fig 8.** PCA1 is a major axis that underlies overall APD prolongation and explains 52% of the variance. PCA2 mainly detects changes in rise time of the AP upstroke with a smaller contribution of APD$_{tri}$ and explains 25% of the variance.

| PC Axis | APD$_{30}$ | APD$_{50}$ | APD$_{80}$ | APD$_{MXR}$ | APD$_{Tri}$ | Rise Time | Stim Delay | Explained Variance Ratio |
|---|---|---|---|---|---|---|---|---|
| 1 | 0.15 | 0.48 | 0.52 | 0.51 | 0.44 | 0.11 | 0.13 | 0.52 |
| 2 | 0.04 | 0.01 | -0.04 | -0.14 | -0.20 | 0.68 | 0.69 | 0.25 |
| 3 | 0.87 | 0.26 | -0.06 | -0.10 | -0.35 | -0.18 | -0.004 | 0.16 |

and experimental data in the same plot). TTX (purple) and flecainide (green) are clearly separated along the PCA1 axis signifying APD prolongation (left and middle scatter plot of Fig 8E) while flecainide (green) and E4031 (blue) are separated along the PCA axis 2 associated with AP upstroke rise time. These two PCA axes explain most of the variance from drug effect (52% and 25% variance by PCA axis 1 and PCA axis 2 respectively for experimental data (Table 3) 79% and 18% variance by PCA axis 1 and PCA axis 2 respectively for simulation data (Table 2)). PCA axis 3 for the experimental data explains minor variance (16%) (right scatter plot of (Fig 8E)); the contribution of PCA axis 3 was insignificant (3%) for the simulation data.

The multi-class logistic regression model was 96% (Table 4) accurate at using the AP metrics to predict which one of the four pharmacological blockers had been applied to the microtissues. For the simulation data, the performance of the multi-class logistic regression model in predicting which ion channel had been blocked (I$_{Ca}$, I$_{Na}$/I$_{NaL}$, I$_{Ks}$, I$_{Kr}$, or I$_{to}$) was similarly high at 99.9996% (0.00006% SD). These results support the claim that ion channel block by chemical compounds can be detected in hiPSC-derived cardiac microtissues using automated measurements of 6 AP metrics.

## Discussion

Accurate assessment of drug-induced proarrhythmic risk is complex because the activity of multiple ion channels can be affected simultaneously by a single drug, complicating the interpretation of the drug's effect on APD prolongation and arrhythmogenesis. In the present study, we investigated whether changes in AP shape due to ion channel block can be used to infer which channel is blocked using both computational simulations and experimental data. Analysis of feature extraction using PCA on computer simulation and experimental data indicates that selective ion channel block can be deduced from the analysis of AP shapes. We

**Table 4. Accuracy of detecting ion channel blocks from AP shape changes from computer simulation data using multi-class logistic regression.** Results are reported as mean (standard deviation). Standard deviations were obtained by repeating the model fitting, hyperparameter tuning, and testing process on different training/test dataset splits as described in Methods.

| Model | Overall Accuracy | I$_{Na}$ Block (TTX) | I$_{Kr}$/I$_{Na}$ Block (Flecainide) | I$_{Ca}$ Block (Nifedipine) | I$_{Kr}$ Block (E-4031) |
|---|---|---|---|---|---|
| Simulation Data | 0.9996 (6e-5) | 0.9994 (2e-4) | n/a | 0.9988 (3e-4) | 0.9999 (5e-5) |
| Experimental Data | 0.95 (0.03) | 0.95 (0.07) | 0.94 (0.08) | 0.96 (0.06) | 0.96 (0.07) |

achieved this by constructing an advanced automated data analysis pipeline that is computationally simple and robust to measure AP shape metrics capable of predicting specific ion channel blocks.

Measuring alterations in multiple ion channels using the voltage clamp technique can be expensive and time-consuming, and the linking of individual ion channel data to AP dynamics and arrhythmia risks is complex. Our PCA feature extractions comparing computer simulation and experimental data found two major directions of AP shape changes (Figs 2D and 8E, respectively): i) prolongation of APD metrics and $APD_{tri}$ and ii) increase of AP upstroke rise time. This analysis indicates that because each ion channel contributes to different phases of the AP waveform, differentiating AP shape changes between drugs may help identify multiple ion channel targets and evaluate proper risks for arrhythmias.

In recent years, open software packages for analyzing voltage data of intact hearts or tissues have been published [36–39]. These packages, however, are optimized for impulse propagation, dispersion of APD, or calcium transient dynamics from the surface of intact hearts to investigate triggered activity, alternans, conduction block, and reentry formation. Cardiotoxicity testing in hiPSC microtissue platforms are faced with different challenges in AP recording. While optical mapping arrhythmia studies in Langendorff perfused hearts often focus on the spatial dynamics of electrical propagation to identify the location of triggered activity and conduction block responsible for reentry formation, cardiotoxicity studies in microtissues focus on detecting small changes in AP parameters to estimate the potential risks of novel drugs in inciting arrhythmias in a dose-dependent manner. As such, a more robust algorithm capable of 1) removing noise while preserving key APs features and 2) sensitively and accurately detecting differences in AP morphologies is necessary for cardiotoxicity microtissue studies. Furthermore, compared to whole heart imaging, data acquisition can benefit from simultaneous recording of APs from multiple microtissues, which requires accurate identification of individual microtissues from a robust automated segmentation approach. Conventional segmentation of optical mapping data employs Otsu's thresholding of fluorescence images, which may work well when imaging intact hearts. However, many *in vitro* cardiotoxicity platforms employ non-adhesive agarose molds to guide the assembly of 3D microtissues, and these agarose hydrogels often absorb voltage dyes to elevate background fluorescence and reduce image contrast. In this study, image segmentation was improved by FFT and minimization of cross entropy thresholding (Fig 4).

Baseline drift from dye bleaching or perfusion flow is another prevalent issue that interferes with proper AP analysis. Traditionally, AP takeoff point and/or peaks are used as reference points for interpolation when estimating baseline drift and quantifying AP metrics. Noisy AP traces may significantly affect the accuracy of these metrics, especially when aggressive Gaussian smoothing operators are employed. Our approach, therefore, relies on an asymmetric least squares fitting to polynomials that requires only a few iterations (n<5, Fig 5), allowing for accurate measurement of AP upstroke, $APD_{30}$, $APD_{50}$, and $APD_{80}$ (Fig 7). The rise time of the AP upstroke is an important indicator for $Na^+$ channel block, and traditionally, the time interval between 10% to 90% peak has been used. We instead provided an alternative computationally simple approach using a MAS algorithm that can reliably measure the takeoff and peak time of AP upstroke and has the advantage of being less dependent on the smoothing factors such as the window size weight parameters (Fig 6). If needed, the non-linear bilateral filtering proposed here can improve signal-to-noise ratio without distorting the sharpness of AP upstroke (Fig 6). This automated data processing pipeline can remove any potential human biases during data analysis and rapidly analyze drug induced changes in AP metrics to more accurately predict arrhythmia risks.

Our experimental recording and data analysis was able to capture the effect of flecainide on AP shape that is similar to a combination of TTX and E4031. Flecainide is known to block cardiac $Na^+$ channels (IC50 = 7.4 μM [45])and hERG channel (IC50 = 4 μM and other $K^+$ channels (IC50 = 10.7 μM [46]). Both TTX (Fig 8A) and flecainide (Fig 8B), a $Na^+$ channel blocker, demonstrated an increase in AP upstroke rise time and stimulation delay. In contrast to TTX, flecainide also prolonged all APD metrics along with $APD_{tri}$, as predicted by the simulation data and the results observed with $I_{Kr}$ block by E4031. Previous studies [23, 67], including one from our group [35], reported that APD triangulation (defined as APD90 –APD$_{50}$) is associated with hERG channel block. These results highlight the ability of AP metrics to identify multiple ion channel blocks from a single compound.

Although in this study we optimized the above algorithms to detect changes in AP metrics from optical mapping data of self-assembling hiPSC-derived 3D cardiac microtissues, the presented algorithms are also applicable when analyzing AP traces from scaffold-based engineered tissues. It is important to note that although our *in vitro* 3D microtissue model was created under scaffold-free conditions, the addition of hCFs in densely compacting microtissues is likely sufficient to endogenously activate these fibroblasts to deposit extracellular matrix (as reported by others [68]), over the seven-day period of culture. These algorithms are also adaptable for scaffold based engineered tissues of larger scales, with minor modifications in window size. Depending on the pixel resolution of the image, size of the engineered tissue, and tissue conduction velocities, the analysis window size will need to be adjusted accordingly. Since our algorithm is relatively insensitive to increases in window sizes, we do not anticipate this to be an issue. Additionally, depending on the contractile force generated by the engineered tissue, blebbistatin, a potent myosin II inhibitor, may be necessary to minimize motion artifacts. Indeed, we have applied these algorithms to reliably quantify APD metrics from engineered cardiac tissues generated in scaffolds composed of a combination of collagen-1, fibrinogen, and thrombin [69].

There are small differences noted between the human AP model with $Ca^{2+}$ block and the hiPSC-derived cardiac microtissue under nifedipine. Nifedipine mainly blocks $Ca^{2+}$ channels, which shortens the APD in both simulation (Fig 2D) and experiments (Fig 8C). However, the computer simulation indicates that the AP upstroke rise time is also affected by $Ca^{2+}$ channel block, whereas our experimental data does not show changes in AP upstroke rise time. This might be due to electrical field stimulation (10 V/cm) during the experiment that can cause a virtual electrode effect (depolarization in one region and hyperpolarization in the opposite region of 3D microtissue, see S2B Fig in S1 Text, and S1 Movie). This causes slight differences in the AP upstroke rise time between depolarized and hyperpolarized regions. Therefore, subtle changes in the AP upstroke rise time under $Ca^{2+}$ channel block may not be detected experimentally. In addition, this observation is likely due to the fact that the immature hiPSC-derived CMs may not fully reflect the ionic currents of adult human AP model. Indeed, despite continued efforts in the field of tissue engineering to improve maturation, hiPSC-CMs have not yet been able to fully recapitulate the electrical and mechanical functional phenotype of adult cardiomyocytes [70]. To address this issue, several groups have adjusted the original human AP model, including alterations to the $Na^+$ kinetics [71–73] which may improve prediction of drug responses in hiPSC-derived cardiac 3D microtissues.

Logistic regression and PCA analyses on experimental data were able to produce highly accurate predictions of identifying $Na^+$, L-type $Ca^{2+}$, and hERG channel blocks despite a small number of data sets ($n$ = 136, Fig 8 and Table 4). The high predictability was also supported by Logistic regression on computer modeling data with larger number of data sets ($n$ = 199,983). This is most likely because the ion channels included in the data sets have very distinct roles (ie, AP upstroke was most strongly affected by $Na^+$ blockade, plateau by L-type $Ca^{2+}$ channels,

and repolarization by hERG channels) and these distinctions were successfully captured by our set of AP metrics. It is possible that logistic regression models would not perform so well at predicting blockades of other ion channels and exchangers that have more complex impacts on cardiac AP. Logistic regression and PCA analyses on the computer modeling data sets suggested that $I_{to}$ and $I_{Ks}$ block produce unique changes in AP metrics, although these changes are smaller in magnitude than the changes observed in $I_{Na}$, $I_{Ca}$, or $I_{Kr}$ block. The PCA fit to simulation data for a block of all five channels did not clearly separate $I_{to}$ and $I_{Ks}$ from other channel blocks due to the small magnitudes of effects from $I_{to}$ and $I_{Ks}$ blocks (S1 Fig in S1 Text) compared to the changes due to block of $I_{Na}$, $I_{Ca}$ and $I_{Kr}$ (Fig 2). A separate PCA on $I_{to}$ and $I_{Ks}$ block simulation data alone can further separate these blocks (S1 Fig in S1 Text). Experimentally, $I_{Ks}$ block is difficult to detect with the current protocol in our hiPSC-derived cardiac microtissues that show only small changes in AP metrics at very high concentration of chromanol 293B (S5 Fig in S1 Text). Since sympathetic stimulation increases $I_{Ks}$, the impact of $I_{Ks}$ on APD is thought to become more significant under sympathetic stimulation [74]. Heart rate is also an important factor that can affect $I_{Ks}$, and the role of $I_{Ks}$ in modulating APD becomes greater at faster pacing cycle length. Further studies will focus on developing a new protocol to detect alteration in $I_{Ks}$ and $I_{to}$ and the associated arrhythmia risks.

Taken together, in this study, we found that while cardiac fibroblasts helped the compaction of self-assembled 3D microtissues, varying hCF content only contributed to minimal changes in APDs. We developed an automated data analysis pipeline that is computationally simple and robust to evaluate eight unique AP metrics capable of capturing specific ion channel blocks. Feature extraction of AP shape using PCA indicates that major ion channel blocks, including $I_{Na}$, $I_{Ca}$, and $I_{Kr}$ blocks, can be deduced from these AP metrics. This study supports the use of hiPSC-CM self-assembled 3D microtissues as a simple and robust *in vitro* cardiotoxicity testing platform that can be leveraged to establish safe human exposure levels.

## Supporting information

**S1 Text. Supplemental information in methods and python script.** This supplemental document includes S1 to S5 Figs and Python routines of segmentation, baseline subtraction, bilinear filtering, rise time and repolarization measurements using moving average subtraction. (DOCX)

**S1 Movie. This movie file shows fluorescence changes during a field stimulation (10 V/cm).** During field stimulation, microtissues show depolarization in the left and hyperpolarization in the right side, which interferes with correct measurements of AP rise time. (MP4)

## Author Contributions

**Conceptualization:** Arvin H. Soepriatna, Allison Navarrete-Welton, Tae Yun Kim, Peter Bronk, Celinda M. Kofron, Ulrike Mende, Kareen L. K. Coulombe.

**Data curation:** Arvin H. Soepriatna, Allison Navarrete-Welton, Tae Yun Kim, Mark C. Daley, Peter Bronk, Kareen L. K. Coulombe, Bum-Rak Choi.

**Formal analysis:** Arvin H. Soepriatna, Allison Navarrete-Welton, Tae Yun Kim, Mark C. Daley, Bum-Rak Choi.

**Funding acquisition:** Ulrike Mende, Kareen L. K. Coulombe.

**Investigation:** Arvin H. Soepriatna, Allison Navarrete-Welton, Mark C. Daley, Ulrike Mende, Kareen L. K. Coulombe, Bum-Rak Choi.

**Methodology:** Arvin H. Soepriatna, Allison Navarrete-Welton, Tae Yun Kim, Mark C. Daley, Peter Bronk, Celinda M. Kofron, Ulrike Mende, Kareen L. K. Coulombe, Bum-Rak Choi.

**Project administration:** Kareen L. K. Coulombe, Bum-Rak Choi.

**Resources:** Kareen L. K. Coulombe, Bum-Rak Choi.

**Software:** Allison Navarrete-Welton, Tae Yun Kim, Peter Bronk, Bum-Rak Choi.

**Supervision:** Ulrike Mende, Kareen L. K. Coulombe, Bum-Rak Choi.

**Validation:** Arvin H. Soepriatna, Allison Navarrete-Welton, Tae Yun Kim, Mark C. Daley, Peter Bronk, Celinda M. Kofron, Bum-Rak Choi.

**Visualization:** Arvin H. Soepriatna, Allison Navarrete-Welton, Tae Yun Kim, Mark C. Daley, Celinda M. Kofron, Bum-Rak Choi.

**Writing – original draft:** Arvin H. Soepriatna, Allison Navarrete-Welton, Mark C. Daley, Kareen L. K. Coulombe, Bum-Rak Choi.

**Writing – review & editing:** Arvin H. Soepriatna, Allison Navarrete-Welton, Mark C. Daley, Peter Bronk, Celinda M. Kofron, Ulrike Mende, Kareen L. K. Coulombe, Bum-Rak Choi.

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
