## [Decision Letter · Decision Letter 0]

17 Oct 2022

PONE-D-22-19009Action Potential Metrics and Automated Data Analysis Pipeline for Cardiotoxicity Testing Using Optically Mapped hiPSC-derived 3D Cardiac MicrotissuesPLOS ONE

Dear Dr. Choi,

Thank you for submitting your manuscript to PLOS ONE. After careful consideration, we feel that it has merit but does not fully meet PLOS ONE’s publication criteria as it currently stands. Therefore, we invite you to submit a revised version of the manuscript that addresses the points raised during the review process.

Please address all comments indicated by the Reviewers.

We look forward to receiving your revised manuscript.

Kind regards,

Elena G. Tolkacheva, PhD

Academic Editor

PLOS ONE

Journal Requirements:

Reviewers' comments:

Reviewer's Responses to Questions

**Comments to the Author**

1. Is the manuscript technically sound, and do the data support the conclusions?

Reviewer #1: Yes

Reviewer #2: Partly

2. Has the statistical analysis been performed appropriately and rigorously? 

Reviewer #1: Yes

Reviewer #2: I Don't Know

3. Have the authors made all data underlying the findings in their manuscript fully available?

Reviewer #1: Yes

Reviewer #2: No

4. Is the manuscript presented in an intelligible fashion and written in standard English?

Reviewer #1: No

Reviewer #2: Yes

5. Review Comments to the Author

Reviewer #1: In this manuscript, Arvin H et al. developed automated data processing algorithms to assess changes in AP properties for cardiotoxicity testing in 3D engineered cardiac microtissues generated from hiPSC-CMs. This simple and robust automated data analysis pipeline for evaluating key AP metrics provides an excellent in vitro cardiotoxicity testing platform for a wide range of environmental and pharmaceutical compounds. Overall, the subject and perspective are novel and valuable for reference. However, to make the review more rigorous and complete, some points must be concerned:

Major comments

1. In this manuscript, 3D organoid is composed by hydrogel aggregation, and whether the established detection system is also suitable for the detection of 3D organoid action potentials in non-hydrogel media or non-conduction media.

2. The layout of the chart can be more optimized and neat . For example, ①-⑤ should be marked in Figure1; A can be directly represented in Figure2 and Fig2.D can be divided into D-G, ect .

3. In the method part, the writing should be standard and paragraphs should be formatted. For example, "Error! Reference source not found”, “ A shows” ,etc.

4. Is the ordinate unit (#) in Figure4?

5. The manuscript needs an in-depth review by grammar and spelling with a native speaker.

Reviewer #2: In this study, the authors perform automated optical mapping and analysis to extract action potential metrics and assess cardio toxicity. Though this is an interesting study and an important endeavor. I have several major comments that must be addressed prior to the acceptance of this manuscript.

Major Comments

1. All code and relevant example data should be provided on GitHub or similar. At present, code snippets are included within a word doc in the supplementary materials of this manuscript. Instead, these functions should be in “.py” files and stored and disseminated on a public GitHub page or similar platform that is designed for sharing code.

2. In addition to sharing the code as “.py” files, the authors should include full information needed to run both the simulations and analysis code. Specifically, this will require choosing a strategy to share the correct python packages (e.g., requirements.txt file, pyproject.toml file) and installation instructions. And, the authors should provide instructions for running both the simulation code and the analysis software, ideally in the form of a brief tutorial.

3. One key element missing from the paper is a discussion + quantitative comparison of how these techniques compare to recent open source alternatives shared in the literature. As one (of many) potential examples that should be used for direct comparison, the open source software KairoSight seems to provide similar functionality (e.g., https://github.com/kairosight/kairosight-2.0 and https://www.frontiersin.org/articles/10.3389/fphys.2021.752940/full). The authors should create a table with results from direct comparisons between their proposed methods and alternatives currently used in the literature.

4. On p. 8 in the section “Multi-class logistic regression and principal components analysis” — I encourage the authors to double check a few things and to make sure that they are clear. Specifically: (a) Was the standard scalar fitted using the whole dataset or the training dataset? (b) Was all hyper parameter tuning done by examining validation data specifically? Based on the description on page 9 line 237 I am concerned that there is overfitting to the small experimental dataset. For reference, k-fold cross validation must still have held out test data if the ith fold is being used for hyper parameter and/or model selection. Providing the datasets + scripts in a tutorial format on GitHub or similar will greatly increase my confidence in this part of the work.

Minor Comments

1. Page 7 line 159: Error! Reference source not found

2. Page 8 line 186: Error! Reference source not found

3. Funding sources listed in the manuscript vs. on the cover page are inconsistent

4. Are all figures and figure panels referenced in the main body of the paper? It is hard to keep track especially with the error in referencing.

5. Many of the figures are pixelated — this should be addressed.

6. The figure color scheme should be adjusted to be friendly to readers with red/green colorblindness.

6. PLOS authors have the option to publish the peer review history of their article (what does this mean?). If published, this will include your full peer review and any attached files.

Reviewer #1: No

Reviewer #2: No

---

## [Author Response · Author response to Decision Letter 0]

4 Dec 2022

We thank reviewers for the insightful reviews and careful consideration of our manuscript. We found both the general and specific comments extremely helpful in making revisions. We have made several key changes to our manuscript based on reviewers’ critique; 1) we added quantitative comparison between traditional signal processing vs. new routines optimized for simultaneous fluorescence recordings from multiple microtissues, 2) high resolution figures were generated, 3) the color scheme of plots were modified to assist readers with colorblindness, 4) the manuscript was reviewed by native speakers and grammatical and spelling mistakes were corrected. Our point-by-point responses are available in the response to reviewers.

---

## [Decision Letter · Decision Letter 1]

28 Dec 2022

Action Potential Metrics and Automated Data Analysis Pipeline for Cardiotoxicity Testing Using Optically Mapped hiPSC-derived 3D Cardiac Microtissues

PONE-D-22-19009R1

Dear Dr. Choi,

We’re pleased to inform you that your manuscript has been judged scientifically suitable for publication and will be formally accepted for publication once it meets all outstanding technical requirements.

Kind regards,

Elena G. Tolkacheva, PhD

Academic Editor

PLOS ONE

Additional Editor Comments (optional):

Reviewers' comments:

Reviewer's Responses to Questions

**Comments to the Author**

1. If the authors have adequately addressed your comments raised in a previous round of review and you feel that this manuscript is now acceptable for publication, you may indicate that here to bypass the “Comments to the Author” section, enter your conflict of interest statement in the “Confidential to Editor” section, and submit your "Accept" recommendation.

Reviewer #1: All comments have been addressed

Reviewer #2: All comments have been addressed

2. Is the manuscript technically sound, and do the data support the conclusions?

Reviewer #1: Yes

Reviewer #2: Partly

3. Has the statistical analysis been performed appropriately and rigorously? 

Reviewer #1: Yes

Reviewer #2: I Don't Know

4. Have the authors made all data underlying the findings in their manuscript fully available?

Reviewer #1: Yes

Reviewer #2: Yes

5. Is the manuscript presented in an intelligible fashion and written in standard English?

Reviewer #1: Yes

Reviewer #2: Yes

6. Review Comments to the Author

Reviewer #1: The authors developed the automated data processiong algorithms to access action potential in 3D EHT from hiPSC-CMs. They have solved the issued in current version.

Reviewer #2: Thank you to the authors for engaging with the peer review process. Thought I cannot access the GitHub link at present, I trust that the authors will make it public + include detailed documentation for their datasets and code as promised. If this is not published in a timely manner I do not support the publication of this manuscript.

7. PLOS authors have the option to publish the peer review history of their article (what does this mean?). If published, this will include your full peer review and any attached files.

Reviewer #1: No

Reviewer #2: No

---

## [Editor Report · Acceptance letter]

17 Jan 2023

PONE-D-22-19009R1 

Action Potential Metrics and Automated Data Analysis Pipeline for Cardiotoxicity Testing Using Optically Mapped hiPSC-derived 3D Cardiac Microtissues 

Dear Dr. Choi:

I'm pleased to inform you that your manuscript has been deemed suitable for publication in PLOS ONE. Congratulations! Your manuscript is now with our production department. 

Kind regards, 

on behalf of

Dr. Elena G. Tolkacheva 

Academic Editor

PLOS ONE